# Teaching Language Models to Critique via Reinforcement Learning

**Zhihui Xie** [* 1]  **Jie Chen** [* 2]  **Liyu Chen** [2]  **Weichao Mao** [2]  **Jingjing Xu** [2]  **Lingpeng Kong** [1]

https://critic-rl.github.io

## Abstract

Teaching large language models (LLMs) to critique and refine their outputs is crucial for building systems that can iteratively improve, yet it is fundamentally limited by the ability to provide *accurate judgments* and *actionable suggestions*. In this work, we study LLM critics for code generation and propose `CTRL`, a framework for `Critic` `Training` via `Reinforcement` `Learning`, which trains a critic model to generate feedback that maximizes correction performance for a fixed generator model without human supervision. Our results demonstrate that critics trained with `CTRL` significantly enhance pass rates and mitigate compounding errors across both base and stronger generator models. Furthermore, we show that these critic models act as accurate generative reward models and enable test-time scaling through iterative critique-revision, achieving up to 106.1% relative improvements across challenging code generation benchmarks.

## 1. Introduction

Recent advances in Large Language Models (LLMs) have sparked interest in their potential for self-improvement through iterative feedback mechanisms (Pan et al., 2023). Methods like Reflexion (Shinn et al., 2024) and Self-Refine (Madaan et al., 2024) demonstrate that LLMs can, in principle, critique their own outputs and generate refined responses. This self-improvement paradigm offers a promising direction toward more autonomous AI systems that can learn from their mistakes.

However, the effectiveness of such self-improvement mechanisms remains challenging in practice. Huang et al. (2023) demonstrate that without appropriate external feedback, such self-improvement loops may lead to performance

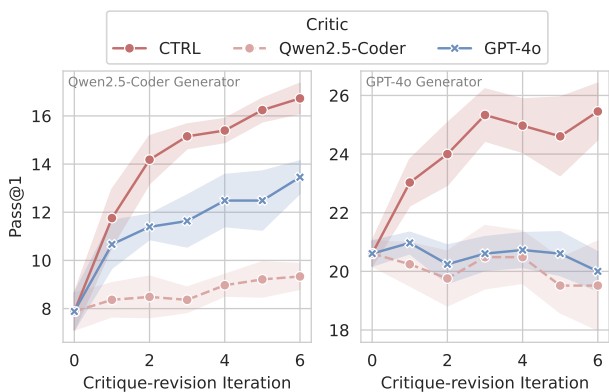

*Figure 1.* Performance scaling of our `CTRL` critic (finetuned on Qwen2.5-Coder-32B-Ins, henceforth Qwen2.5-Coder) compared to other critics across different generators on CodeContests. `CTRL` demonstrates strong critiquing capabilities not only when paired with its base model but also with a stronger generator (GPT-4o, right). Shaded regions indicate standard error across 5 seeds.

degradation. To address this, existing approaches primarily rely on reward models (Sun et al., 2023; Yuan et al., 2024) or automated verification tools (Gou et al., 2023; Chen et al., 2023). However, these mechanisms often fail to provide actionable guidance — reward models compress complex evaluation criteria into simplified numerical signals (Gao et al., 2023; Pan et al., 2024), while verification tools generate low-level execution traces that do not directly translate to high-level fixes (Zhong et al., 2024). Even in domains like code generation (Li et al., 2022; Sun et al., 2024) where such feedback mechanisms are readily available, previous work (Zheng et al., 2024) as well as our experiment (Table 1) reveal that such feedback alone struggles to drive meaningful improvements. At the heart of this issue lies the *feedback bottleneck*: feedback needs to both accurately discriminate the correctness of solutions and provide informative yet actionable suggestions for improvement.

To address these challenges, we propose `CTRL` (`Critic` `Training` via `Reinforcement` `Learning`), a framework that decouples the critic model from the task-performing model (e.g., GPT-4o) and focus on developing a specialized critic that can effectively drive the task-performing model toward optimal solution generation through iterative critique-revisions (Figure 2). This decomposition naturally introduces a well-defined *proxy task* for training the critic model:

[*]Equal contribution  [1]The University of Hong Kong  [2]Bytedance Seed. Correspondence to: Zhihui Xie <zhxieml@gmail.com>.

*Proceedings of the 42nd International Conference on Machine Learning*, Vancouver, Canada. PMLR 267, 2025. Copyright 2025 by the author(s).

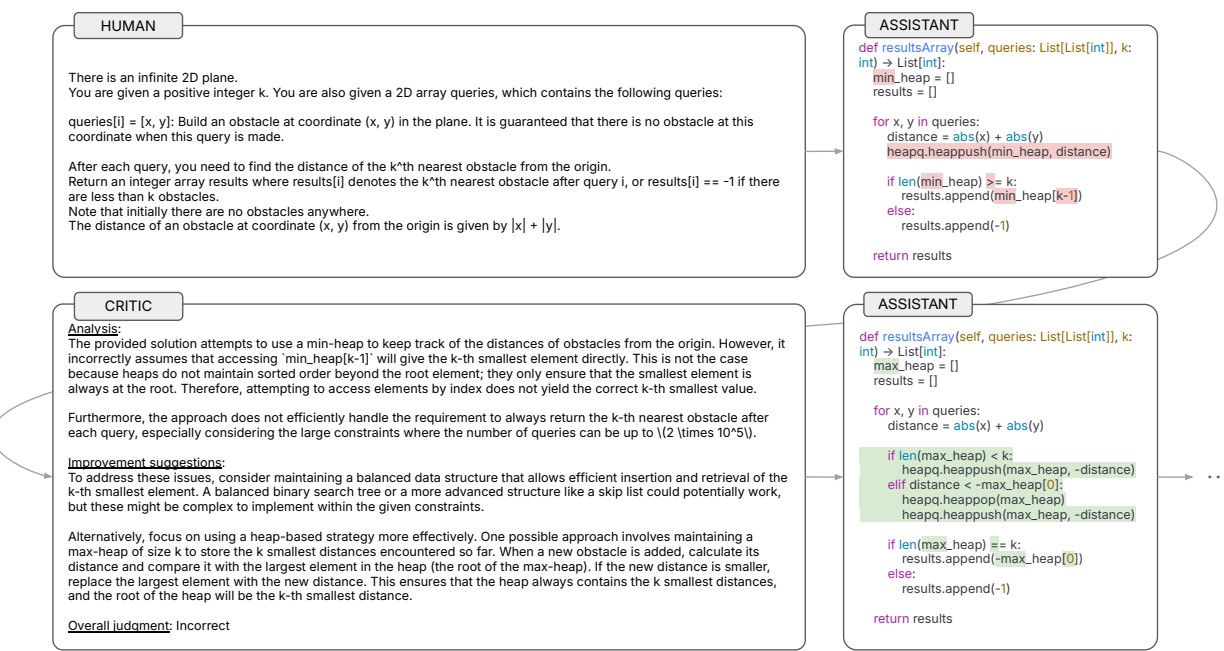

*Figure 2.* Illustration of the critique-correction process for a coding problem. Top: An initial solution is proposed by the task-performing using a min-heap approach. Bottom: The critic identifies flaws in the implementation (incorrect heap access and inefficient query handling) and suggests specific improvements, leading to a corrected max-heap solution. This example is taken from critiques of CTRL on LiveCodeBench, which demonstrates how structured feedback from the critic can guide meaningful improvements in code generation.

while directly evaluating the quality of generated critiques remains challenging, the effectiveness of a critic can be measured by its ability to drive the task-performing model toward correct outputs. Though such indirect optimization signals lead to a large space of possible critiques and therefore high variance during training, we address this through a two-stage pipeline: first synthesizing high-quality critiques using execution feedback for supervised finetuning, then optimizing the critic through Group Relative Policy Optimization (GRPO; Shao et al. 2024).

Through extensive evaluations on diverse benchmarks including CodeContests (Li et al., 2022), LiveCodeBench (Jain et al., 2024), MBPP+ (Liu et al., 2024a), and JudgeBench (Tan et al., 2024), we demonstrate that training with CTRL significantly outperforms both self-critique approaches and methods using stronger critic models. Notably, we observe remarkable generalization capabilities of the decoupled critic LLM across different problem domains and model scales. Our experiments demonstrate that relatively weaker critic models can effectively guide stronger task-performing models such as GPT-4o (Table 3), exhibiting a similar phenomenon to weak-to-strong generalization (Christiano et al., 2018; Burns et al., 2023), where weaker models can be trained to effectively supervise more capable ones.

Furthermore, CTRL enables efficient test-time scaling (Figure 1). By providing targeted and actionable feedback, our critic significantly reduces the number of revision iterations needed, leading to both lower token consumption and higher success rates. Our empirical analysis (Figure 4) demonstrates that this efficiency stems from reduced error compounding—the critic effectively identifies and corrects mistakes early, guiding the model toward more direct solution paths without compromising solution quality.

Our work makes four key contributions: (1) We propose CTRL, a novel framework that decouples critic LLMs from task-performing models and trains them through two-stage GRPO to guide code improvement. (2) Through extensive evaluation on programming benchmarks, we demonstrate that CTRL significantly outperforms both self-critique methods and approaches using stronger critic models. (3) We establish that relatively weaker critic models can effectively guide stronger task-performing models, demonstrating a promising weak-to-strong generalization phenomenon in LLM guidance. (4) We show that a trained critic enables test-time scaling through iterative critique-revisions, achieving up to 106.1% and 23.5% relative Pass@1 improvements on the challenging CodeContests benchmark when paired with its base model and a stronger model, respectively.

## 2. Preliminaries and Motivation

The success of iterative improvement methods critically depends on their ability to leverage feedback to improve

solutions. Formally, let $x$ be an input problem and $y$ be a candidate solution, with $R(y)$ being the evaluation function that returns 1 if $y$ is correct and 0 otherwise. Starting with an initial proposal distribution $y_0 \sim \pi(\cdot \mid x)$, the iterative process generates subsequent solutions by incorporating feedback $f(\cdot \mid x, y_i)$ and produce the next solution $y_{i+1}$.

In this context, the effectiveness of such feedback mechanisms relies on two key capabilities: (1) *discrimination* - the ability to evaluate and rank solutions, and (2) *critiquing* - the ability to provide actionable feedback for improvement. While discrimination has been extensively studied (Gao et al., 2023), we focus on the critiquing ability and propose to characterize it through the transition dynamics of a Markov chain (Meyn & Tweedie, 2012) governing the correctness of the iteratively refined solutions $\{R(y_i)\}_i$:

$$P(R(y_0) = 1) = p_{\text{init}},$$
$$P(R(y_{i+1}) = 1 \mid R(y_i) = 1) = p_{\text{cc}},$$
$$P(R(y_{i+1}) = 1 \mid R(y_i) = 0) = p_{\text{cw}},$$

where $p_{\text{cc}}$ represents the critiquing ability to avoid turning correct solutions into wrong ones, and $p_{\text{cw}}$ captures the helpfulness of the feedback in improving the solution.

**Varying Critiquing Ability.** To understand the importance of the critiquing ability, we conduct simulations across different levels of critiquing strength while leveraging discrimination to aggregate the final solutions. We consider $p_{\text{init}} = 0.1$ and three scenarios: (1) No critiquing ($p_{\text{cw}} = p_{\text{cc}}$), a special case representing methods that independently sample from the base distribution, or equivalently best-of-$n$ sampling (Sessa et al., 2024); (2) Weak critiquing ($p_{\text{cc}} = 0.7$, $p_{\text{cw}} = 0.15$); and (3) Strong critiquing ($p_{\text{cc}} = 0.9$, $p_{\text{cw}} = 0.3$). For each scenario, we first generate $n$ solutions based on the specified transition dynamics, then apply the discrimination ability to select the best promising solution, and plot the final correctness probability against the number of attempts $n$. We present more details in Appendix C.1.

**Observations & Takeaways.** As shown in Figure 3, our analysis reveals several key findings: **(1)** Strong critiquing abilities significantly improve success rates compared to no critiquing, with performance gains visible even with weak critiquing, aligning with recent empirical findings (Huang et al., 2023). **(2)** Strong critiquing ability can compensate for weaker discrimination — a system with weak discrimination but strong critiquing feedback can outperform one with stronger discrimination but no critiquing ability. **(3)** The benefits of critiquing compound with more iterations, while approaches with no critiquing plateau quickly. These findings highlight that effective iterative improvement requires careful attention to both discrimination and critiquing

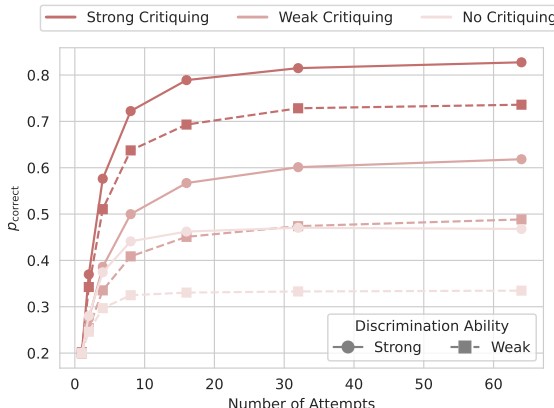

*Figure 3.* Simulation results showing success probability ($p_{\text{correct}}$) as a function of the number of attempts, comparing different levels of critiquing and discrimination ability.

abilities. While perfect abilities are not necessary, systematically improving these capabilities — particularly the ability to generate actionable critiques — is crucial for realizing the full potential of iterative refinement approaches.

## 3. Method

With analysis presented in Section 2, our goal is to teach LLMs the ability of critiquing without human supervision. We propose CTRL, a two-stage training approach: (1) synthesizing high-quality critiques by reasoning about execution feedback, then (2) refining the critic through reinforcement learning. Once trained, the critic model can be used at test time, paired with any generator models, to iteratively refine solutions. A complete overview of the pipeline is provided in Appendix A, with critique samples in Appendix E.

### 3.1. Problem Statement

We focus on code generation as our primary domain as it provides clear objective metrics through test cases, following previous work (McAleese et al., 2024). Given a programming problem $x$ (specified in natural language) and a solution $y$ (code implementation), our goal is to enable iterative refinement of solutions, which centers on two key components: (1) a generator model $\pi(y \mid x)$ that proposes solutions, and (2) a critic model $C_\theta(c|x, y)$ that provides textural feedback $c$ for improvement.

**Assumptions.** Let $\mathcal{D} = \{(x_i, T_i)\}_{i=1}^N$ be our training dataset, where each problem $x_i$ is paired with unit tests $T_i$. We have access to a sandbox environment that executes code against test cases, which serves as the evaluation function $R(y)$ that returns 1 if $y$ passes all tests, 0 otherwise. Notably, the sandbox does not assist critique generation at test time. While not required, we treat the generator model as a *black-box*, allowing our approach to build upon existing

strong generators without access to their parameters.

**Objective.** While directly measuring the helpfulness of critiques remain challenging, we can define a *proxy task* that evaluates whether the critique leads to improved solutions. Given an initial solution $y' \sim \pi(\cdot \mid x)$, the critic analyzes it and produces textual feedback $c$. The generator then uses this feedback to revise the solution, producing an improved output $y$. Let $z = (x, y')$ represent the problem-solution pair. Our objective is to train the critic model $C_\theta$ to maximize the expected solution quality:

$$\mathcal{J}(\theta) = \mathbb{E}_{z \sim \mathcal{D} \times \pi, y \sim \pi_\theta(\cdot|z)}[R(y)], \quad (1)$$

where $\pi_\theta(y \mid z) = \sum_c C_\theta(c \mid z)\pi(y \mid z, c)$ denotes the improved solution distribution through marginalization over possible critiques. Notably, although Equation (1) defines a single-turn critique-revision task, we observe that the trained model generalizes to multi-turn revisions (Section 4.2).

**Defining the Critique Space.** We structure the critique space into three components (Figure 2): (1) an analysis of the solution's strengths and weaknesses, (2) actionable improvement suggestions, and (3) a final judgment of correctness (correct/incorrect). During inference, these components enable iterative critique-revision, where the process stops once the judgment indicates the solution is correct. This design balances discrimination and critiquing, both essential for iterative refinement, as discussed in Section 2.

### 3.2. Stage I: Execution-guided Critique Synthesis

Although conceptually straightforward, learning effective critiques is challenging due to the large critique space, where only a small fraction leads to successful revisions. Our experiments with Qwen2.5-Coder (Hui et al., 2024) (Table 1) show that models struggle to generate informative critiques for self-improvement, aligning with previous findings (Huang et al., 2023). Self-critique without additional feedback yields minimal gains (7.88% → 8.36%) and rarely converts incorrect solutions to correct ones, highlighting the limited ability of models to correct their own mistakes.

**Reasoning over Execution.** While the initial critiquing ability is limited, previous work (Ni et al., 2024) has shown that LLMs can effectively reason over execution feedback. Table 1 demonstrates that when LLMs reason over execution feedback to generate critiques (Self-critique w/ Execution Feedback), they achieve substantial improvements, as compared to directly using raw execution feedback for revisions (11.76% vs. 8.97%). This suggests that while directly using raw execution feedback is inefficient, we can leverage the model's reasoning ability over execution feedback to help generate more accurate and informative critiques.

*Table 1.* Critique-revision performance (Pass@1, %) on CodeContests. We fix the generator model to be Qwen2.5-Coder, and compare zer-shot performance with critique-revision performance using different feedback mechanisms. $\times k$ represents conducting iterative critique-revision $k$ times. $^\dagger$using unit tests for generation.

| | **Pass@1** | $\Delta_\uparrow$ | $\Delta_\downarrow$ |
|---|---|---|---|
| Zero-shot | 7.88 | 0.00 | 0.00 |
| Execution Feedback (EF)$^\dagger$ | 8.97 | 2.42 | 1.33 |
| Self-critique w/ EF$^\dagger$ | 11.76 | 3.88 | **0.00** |
| Self-critique | 8.36 | 2.30 | 1.82 |
| Critique w/ CTRL$_{\text{SFT}}$ | 8.36 | 3.52 | 3.03 |
| Critique w/ CTRL | 11.76 | 4.73 | 0.85 |
| Critique$\times$2 w/ CTRL | 14.18 | 7.27 | 0.97 |
| Critique$\times$3 w/ CTRL | **15.15** | **8.12** | 0.85 |

*Table 2.* Discrimination performance (F1 score, %) on CodeContests.

| | **Passed** | **Failed** | **Macro** |
|---|---|---|---|
| Qwen2.5-Coder | 88.21 | 34.16 | 61.19 |
| CTRL$_{\text{SFT}}$ | **95.54** | 41.26 | 68.55 |
| CTRL | 93.19 | **45.02** | **69.10** |

**Critique Synthesis.** Building on the above insight, we develop a critique synthesis approach that leverages execution feedback to train models in generating effective critiques. Our approach samples high-quality synthesized critiques from a hinted distribution $C_\theta(c \mid z, h)$, where hints $h$ are constructed by analyzing initial solutions $y'$ through sandbox execution. We map different execution outcomes to specific hint templates as shown in Table 7: (1) for passing solutions, we encourage concise positive feedback; (2) for completely failing solutions, we suggest restarting from scratch; and (3) for partially failing solutions, we provide the exact error message and test case details to help pinpoint the issue.

**Supervised Finetuning.** Similar to context distillation (Snell et al., 2022; Guan et al., 2024), we exclude these hints and conduct supervised finetuning to encourage the model to internalize the critiquing strategies. We observe leveraging execution feedback for supervised finetuning is beneficial mainly in two aspects: (1) it helps learn the format; (2) while it marginally improves the critique-revision performance due to the high frequency of instructing correct solutions to wrong (Table 1), it substantially boosts discrimination by providing ground-truth correctness (Table 2).

### 3.3. Stage II: Reinforced Critique Generation

While our critique synthesis approach with predefined templates provides a strong foundation, it may not capture all nuanced feedback scenarios required for complex programming tasks. To overcome this limitation, we formulate

critique generation as a reinforcement learning problem, allowing the critic to adaptively learn feedback strategies through direct optimization of solution improvement.

Our goal is to maximize the performance in Equation (1). To optimize $C_\theta$, one natural approach is using policy gradient methods (Sutton et al., 1999):

$$
\begin{aligned}
&\nabla_\theta \mathbb{E}_{y \sim \pi_\theta}[R(y)] \\
=&\nabla_\theta \mathbb{E}_{y \sim \sum_c C_\theta(c|z)\pi(y|z,c)}[R(y)] \\
=&\nabla_\theta \sum_y R(y) \sum_c C_\theta(c|z)\pi(y|z,c) \\
=&\sum_y R(y) \sum_c \nabla_\theta C_\theta(c|z)\pi(y|z,c)
\end{aligned}
$$

The double summation over both solution space $y$ and feedback space $c$ introduces high variance in gradient estimates:

$$
\mathrm{Var}(\nabla_\theta) = \mathbb{E}[(\nabla_\theta - \mathbb{E}[\nabla_\theta])^2] \propto |\mathcal{Y}| \cdot |\mathcal{C}|.
$$

where $|\mathcal{Y}|$ and $|\mathcal{C}|$ are the sizes of solution and critique spaces respectively. In this scenario, using value networks to predict credit assignment remains challenging, as we observe significant instability when using Proximal Policy Optimization (PPO; Schulman et al. 2017) — the learned networks produce noisy estimates of critique quality. We present detailed experimental observations in Appendix D.

**Variance Reduction.** To combat these variance issues, we adopt Group Relative Policy Optimization (GRPO; Shao et al. 2024) that avoids using value networks for learning credit assignment and reduces variance through group-based relative advantages. Specifically, for each problem-solution pair $z = (x, y')$, we sample a group of critiques $\{c_1, c_2, ..., c_G\}$ from $C_\theta(\cdot|z)$ and compute advantages:

$$
A_i = \frac{R(y_i) - \mu_G}{\sigma_G},
$$

where $y_i \sim \pi(\cdot|z, c_i)$ is the improved solution generated using critique $c_i$, and $\mu_G$ and $\sigma_G$ are the mean and standard deviation of rewards within the group. This approach normalizes rewards across different problem types and naturally focuses training on problems where critique quality can make a meaningful difference, as problems that are too easy or too hard produce zero relative advantages. The final training objective is:

$$
\begin{aligned}
\mathcal{J}(\theta) = \mathbb{E}_{z \sim \mathcal{D}, \{c_i\}_{i=1}^G \sim C_{\theta_{old}}(\cdot|z)} \Big[ \\
\frac{1}{G} \sum_{i=1}^G \Big( \min \big( \frac{C_\theta(c_i|z)}{C_{\theta_{old}}(c_i|z)} A_i, \mathrm{clip}_\varepsilon \big( \frac{C_\theta(c_i|z)}{C_{\theta_{old}}(c_i|z)} \big) A_i \big) \Big) \\
- \beta \mathbb{D}_{\mathrm{KL}}(C_\theta \| C_{\mathrm{ref}}) \Big],
\end{aligned}
$$

where $\mathrm{clip}_\varepsilon$ represents clipping the value to $[1 - \varepsilon, 1 + \varepsilon]$ and $\mathbb{D}_{\mathrm{KL}}(C_\theta \| C_{\mathrm{ref}}) = \frac{C_{\mathrm{ref}}(c_i|z)}{C_\theta(c_i|z)} - \log \frac{C_{\mathrm{ref}}(c_i|z)}{C_\theta(c_i|z)} - 1$ denotes the KL regularization term that alleviates over-optimization.

# 4. Experiments

We conduct extensive experiments to evaluate our method's effectiveness across multiple benchmarks. Our evaluation focuses on two key aspects: (1) the accuracy of the critic in identifying solution correctness, and (2) the quality improvement achieved through critique-guided revisions.

## 4.1. Setup

**Training Data.** We use TACO (Li et al., 2023), a dataset containing 26,443 programming problems collected from competitive programming platforms like CodeForces and LeetCode. Each problem includes a natural language description and multiple test cases. Due to noise in the original dataset (malformed test cases and contaminated problems), we filter the dataset to 18,820 problems for training, with details presented in Appendix C.3.

**Models.** We base our critic model on the open-source Qwen2.5-Coder-Ins (Hui et al., 2024) model. During training, we fix the generator model to be Qwen2.5-Coder-Ins itself. For evaluation, we assess the trained critic's performance by pairing it with various generator models for initial solution generation and subsequent revision, comparing against other LLM critics such as GPT-4o.

**Benchmarks.** We evaluate our approach on three programming benchmarks and one general-domain benchmark: (1) CodeContests (Li et al., 2022), a collection of challenging competitive programming problems; (2) LiveCodeBench (24.08-24.11) (Jain et al., 2024), a curated set of recent programming challenges designed to minimize data contamination; (3) MBPP+ (Liu et al., 2024a), an extension of the MBPP benchmark (Austin et al., 2021) focused on fundamental programming tasks; and (4) JudgeBench (Tan et al., 2024), where we evaluate the model's effectiveness as a generative reward model for comparing solution pairs.

**Metrics.** To evaluate critiquing ability, we use three metrics: Pass@1 measures the success rate of the final solutions, $\Delta_\uparrow$ represents the fraction of initially incorrect solutions that become correct after revision, and $\Delta_\downarrow$ represents the fraction of initially correct solutions that become incorrect after revision. For discrimination ability, we employ F1 score when evaluating single solutions, and accuracy when comparing paired solutions in Judgebench, as the latter involves binary decisions between two alternatives.

*Table 3.* Performance comparison across different generators and benchmarks. We evaluate different configurations, with critique-revision representing an iterative process where a critic model provides feedback to guide solution improvement. Pass@1 shows the success rate, while $\Delta_\uparrow$ and $\Delta_\downarrow$ indicate the percentage of wrong solutions being correctly revised and correct solutions being revised to wrong solutions, respectively. Results are averaged over 5 random seeds.

| | CodeContests | | | LiveCodeBench | | | MBPP+ | | | Average |
| | Pass@1 | $\Delta_\uparrow$ | $\Delta_\downarrow$ | Pass@1 | $\Delta_\uparrow$ | $\Delta_\downarrow$ | Pass@1 | $\Delta_\uparrow$ | $\Delta_\downarrow$ | Pass@1 |
|---|---|---|---|---|---|---|---|---|---|---|
| *Qwen2.5-Coder as Generator* | | | | | | | | | | |
| Zero-shot | 7.88 | 0.00 | 0.00 | 30.54 | 0.00 | 0.00 | 77.83 | 0.00 | 0.00 | 38.75 |
| *Single-turn Critique-revision* | | | | | | | | | | |
| Critique w/ Qwen2.5-Coder | 8.36 | 2.30 | 1.82 | 32.14 | 2.50 | 0.89 | 77.83 | 3.49 | 3.49 | 39.45 |
| Critique w/ GPT-4o | 10.67 | **4.85** | 2.06 | 32.32 | 2.32 | **0.54** | 77.46 | **3.81** | 4.18 | 40.15 |
| Critique w/ CTRL | **11.76** | 4.73 | **0.85** | **33.21** | 3.39 | 0.71 | **78.84** | 2.43 | **1.43** | **41.27** |
| *Multi-turn Critique-revision* | | | | | | | | | | |
| Critique×5 w/ Qwen2.5-Coder | 9.21 | 3.76 | 2.42 | 29.64 | 2.14 | 3.04 | 76.03 | 3.81 | 5.61 | 38.30 |
| Critique×5 w/ GPT-4o | 12.48 | 7.03 | 2.42 | 32.86 | **4.82** | 2.50 | 74.60 | **4.34** | 7.57 | 39.98 |
| Critique×5 w/ CTRL | **16.24** | 9.21 | **0.85** | **33.39** | 3.75 | **0.89** | **78.68** | 3.23 | 2.38 | **42.77** |
| *GPT-4o as Generator* | | | | | | | | | | |
| Zero-shot | 20.61 | 0.00 | 0.00 | 32.32 | 0.00 | 0.00 | 77.67 | 0.00 | 0.00 | 43.53 |
| *Single-turn Critique-revision* | | | | | | | | | | |
| Critique w/ Qwen2.5-Coder | 20.24 | 3.52 | 3.88 | **35.36** | **3.93** | 0.89 | 76.67 | 0.85 | 1.85 | 44.09 |
| Critique w/ GPT-4o | 20.97 | 2.30 | **1.94** | 34.82 | 2.68 | **0.18** | 77.41 | **1.01** | 1.27 | 44.40 |
| Critique w/ CTRL | **23.03** | 4.97 | 2.55 | 33.39 | 2.14 | 1.07 | **77.83** | 0.53 | **0.37** | **44.75** |
| *Multi-turn Critique-revision* | | | | | | | | | | |
| Critique×5 w/ Qwen2.5-Coder | 19.52 | 5.21 | 6.30 | **35.54** | **5.36** | 2.14 | 76.08 | 1.53 | 3.12 | 43.71 |
| Critique×5 w/ GPT-4o | 20.61 | 3.39 | 3.39 | 35.18 | 3.21 | **0.36** | 76.61 | **2.06** | 3.12 | 44.13 |
| Critique×5 w/ CTRL | **25.45** | 7.88 | **3.03** | 34.11 | 3.21 | 1.43 | **77.94** | 0.79 | **0.53** | **45.83** |

**Execution Sandbox.** We employ SandboxFusion (Liu et al., 2024b) as our execution environment, which provides a unified interface for evaluating solutions across training data and benchmarks through both function-based and standard input-output formats.

### 4.2. Evaluating Critics for Iterative Critique-revisions

To evaluate the effectiveness of CTRL, we present a comprehensive analysis of critique-revision strategies with different feedback mechanisms on CodeContests in Table 1. The discrimination performance of critics is shown in Table 2, while results across different benchmarks and generators are presented in Table 3.

**RL Significantly Boosts Critiquing Ability.** Table 1 shows that our RL-trained critic significantly outperforms baseline approaches, achieving a 11.76% pass@1 rate compared to 7.88% with zero-shot generation. This substantial improvement builds upon a much reduced regression rate $\Delta_\downarrow$ than its SFT counterpart (0.85% vs. 3.03%).

**CTRL Enables Test-time Scaling.** As shown in Table 1, our approach enables test-time scaling through iterative critique-revisions. Notably, despite training exclusively on single-turn critiquing tasks, CTRL generalizes to multi-turn settings. By increasing the number of iterations from one to three (Critique×3 w/ CTRL), we further improve the

Pass@1 rate from 11.76% to 15.15% while maintaining a low regression rate $\Delta_\downarrow$ of 0.85%. This demonstrates that our critic provides consistently reliable feedback across multiple revision iterations, unlike baseline approaches that accumulate errors, as discussed below.

**CTRL Mitigates Compounding Errors.** Figure 4 further illustrates this stability advantage - while both Qwen2.5-Coder and GPT-4o show increasing error compounding rates over iterations, CTRL maintains a significantly lower rate, enabling reliable multi-round improvements.

**CTRL Generalizes to Different Generators and Tasks.** While we train the critic model with Qwen2.5-Coder as the generator, as shown in Table 3, our approach generalizes well across different programming tasks. Notably, a weak critic model trained against itself can assist stronger model (GPT-4o), providing evidence for scalable oversight (Christiano et al., 2018; Kenton et al., 2024).

**Performance Scaling with Problem Difficulty.** As shown in Figure 5, our critique-revision approach demonstrates increasingly substantial relative gains as both iteration and problem difficulty increases, revealing that CTRL is particularly effective for complex tasks, where iterative refinement through targeted critique and revision yields the most significant benefits compared to zero-shot generation.

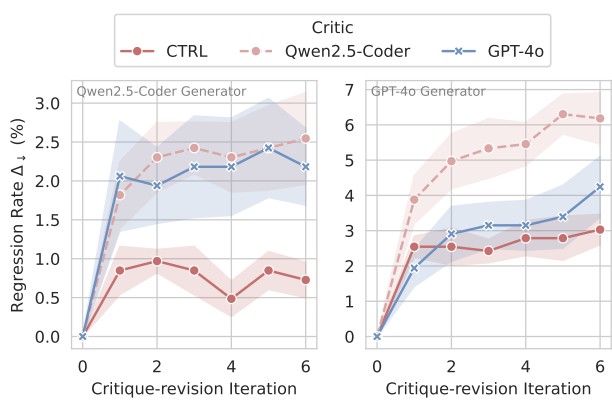

Figure 4. Compounding error analysis. Regression rate measures the frequency of correct initial solutions being revised into incorrect ones. Shaded regions indicate standard error over 5 seeds.

### 4.3. Evaluating Critics as Generative Reward Models

One advantage of unifying textural feedback is to balance discrimination and critiquing abilities. To assess our critics' discrimination capabilities, we evaluate them on JudgeBench (Tan et al., 2024), a comprehensive benchmark containing 350 GPT-4o completions across categories spanning general knowledge, reasoning, mathematics, and coding. This setup presents a challenging out-of-distribution test in two aspects: **(1)** our critics must evaluate outputs from a more capable model than their training distribution, and **(2)** they need to generalize to broader domains beyond coding tasks. This evaluation scenario is particularly interesting as it examines whether relatively weaker models can be effectively trained to judge outputs from more powerful models.

As shown in Figure 6, `CTRL` critic achieves competitive performance compared to stronger models such as Claude-3.5-Sonnet. Notably, while our critic is specifically trained on programming tasks, it maintains comparable overall accuracy (64.3%) while demonstrating superior performance

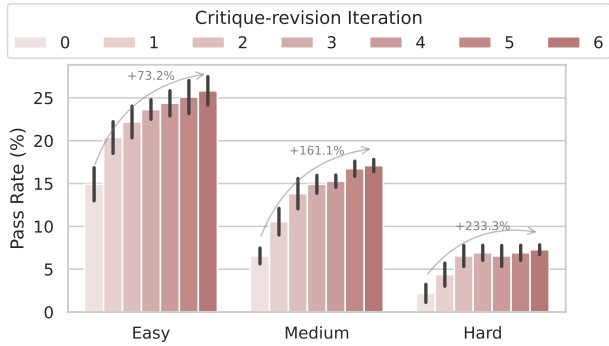

Figure 5. Comparison of pass@1 rates by problem difficulty with `CTRL` critics on CodeContests. Results are averaged over 5 seeds.

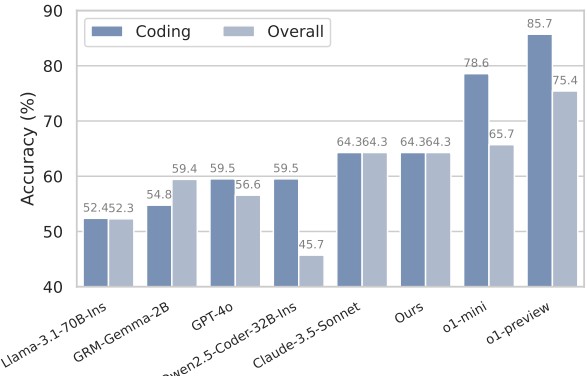

Figure 6. Model performance comparison on JudgeBench.

on coding-specific evaluations. This suggests that our `CTRL` enables effective discrimination capabilities that generalize beyond the training domain.

### 4.4. Analysis

To better understand how `CTRL` boosts iterative refinement, we further conduct analyses on the similarity between original and revised solutions, execution time changes, and critique characteristics. Our findings reveal several key patterns in how different critique methods influence the process of critique-revision.

**The Effect of Generator Ability.** As a preliminary analysis before finetuning experiments, we examine how model sizes affect critique-revision performance using Qwen2.5-Coder-Ins models (7B, 14B, and 32B) in an *inference-only* setting, comparing zero-shot generation against critique-revision with critiques generated by another critic model conditioned on execution feedback. Table 4 reveals that critic capability significantly influences improvement potential—while smaller critics (7B) often lead to performance degradation, larger critics (32B) consistently yield better outcomes, achieving up to 50% improvement when paired with

Table 4. Relative improvement (%) on CodeContests when comparing critique-revision (using critics conditioned on execution feedback) against zero-shot generation, across different generator-critic size combinations. Results are from inference-only experiments before any finetuning.

| Generator | Critic | | | Avg. |
| | 7B | 14B | 32B | |
|---|---|---|---|---|
| 7B | -33.33 | 22.22 | -11.11 | -7.41 |
| 14B | -9.09 | -9.09 | 9.09 | -3.03 |
| 32B | 0.00 | 30.00 | 50.00 | **26.67** |
| Avg. | -14.14 | 14.38 | **15.99** | |

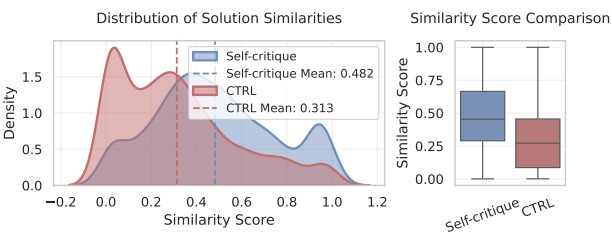

*Figure 7.* Comparison of solution similarities between original and revised code guided by CTRL on CodeContests. Left: Distribution of similarity scores for self-critique and our CTRL method. Right: Box plot showing the statistical distribution of similarity scores. Lower scores indicate more substantial revisions.

similarly-sized generators. The results also highlight the importance of critic-generator size relationships, as critics less capable than their generators typically degrade performance. These findings motivate us to focus our subsequent finetuning experiments with CTRL on 32B models to maximize the benefits of critique-revision.

**CTRL Prevents Similar Revisions.** We analyze how different critique methods influence solution revisions by measuring code similarity scores between original and revised solutions, as described in Appendix C.4. As shown in Figure 7, self-critique tends to make conservative modifications with higher similarity scores (mean 0.482), while our CTRL method proposes more substantial changes (mean 0.313). This suggests CTRL is more willing to recommend major structural revisions when needed, rather than just local optimizations, which may explain its superior performance in improving solution quality.

**CTRL Trade-offs between Accuracy and Efficiency.** While our critique-revision approach improves solution accuracy on LiveCodeBench, we observe a notable increase in timeout rates. Solutions guided by CTRL exhibit a timeout rate of 16.61%, higher than both zero-shot (10.54%) and GPT-4o critic (8.93%). However, even with more timeouts, CTRL still achieves better overall Pass@1 accuracy. This suggests that our approach tends to generate more comprehensive solutions — while these may take longer to execute, the solution quality is guaranteed.

## 5. Related Work

**Self-Improvement of LLMs.** Recent work has explored various approaches for LLMs to improve their outputs autonomously, including self-reflection (Shinn et al., 2023; Feng et al., 2024), self-critique (Madaan et al., 2024; Shinn et al., 2024), debates (Irving et al., 2018; Michael et al., 2023; Khan et al., 2024), and training models to self-correct (Welleck et al., 2022; Kumar et al., 2024). However,

*Table 5.* Timeout rate and Pass@1 (%) on LiveCodeBench. While CTRL approach achieves higher pass rates, it tends to generate more comprehensive solutions that take longer to execute.

|  | **Timeout Rate** ($\downarrow$) | **Pass@1** ($\uparrow$) |
| --- | --- | --- |
| Zero-shot | 10.54 | 30.54 |
| Critique w/ GPT-4o | **8.93** | 32.32 |
| Critique w/ CTRL | 16.61 | **33.21** |

Huang et al. (2023) demonstrates that without appropriate external feedback, such self-improvement loops may lead to performance degradation. Our work addresses these challenges by learning specialized models that can provide effective feedback for improvement.

**LLM Critics.** Several approaches have been proposed to train LLMs as critics for various purposes, including generative reward models (Ankner et al., 2024; Xiong et al., 2024) and scalable oversight (Saunders et al., 2022; Kenton et al., 2024). These approaches either learn from human feedback (Wang et al., 2023; McAleese et al., 2024) or much more capable models' outputs (Xi et al., 2024), with recent work exploring reinforcement learning to improve feedback generation (Akyürek et al., 2023; Yao et al., 2023). Our approach differs in three key aspects: (1) leveraging execution feedback and model reasoning to synthesize high-quality critiques, (2) introducing variance reduction techniques to stabilize training, and (3) requiring only single-round critique-revision interactions. Additional discussion on related work is provided in Appendix B.

**Scaling Test-Time Compute.** Recent work has explored various approaches to improve model performance at test time without fine-tuning (Snell et al., 2024). While existing approaches focus on techniques like repeated sampling with proper selection mechanisms (Brown et al., 2024) and more sophisticated modular frameworks with existing models (Saad-Falcon et al., 2024), we instead investigate test-time scaling through a decoupled critic model trained to provides targeted feedback to guide solution improvements. Notably, while Saad-Falcon et al. (2024) demonstrates that strong models can serve as effective critics, their approach struggles with code generation tasks.

## 6. Conclusion

We present CTRL, a reinforcement learning framework for training critic LLMs to provide effective feedback for iterative refinement. Our trained critic demonstrates significant improvements over baselines across multiple benchmarks and enables efficient test-time scaling through iterative critique-revisions — notably, even when guiding stronger generators. While this work focuses on improving pass

rates, future directions include optimizing for efficiency and safety, and extending our training pipeline towards multi-turn critique revision. We hope this work inspires further research into scalable LLM self-improvement through reinforcement learning.

## Impact Statement

This work aims to advance the field of Machine Learning by introducing a framework for training LLM critics. While this research has the potential to improve the reliability and robustness of AI systems, we have not identified any immediate societal concerns requiring specific attention. However, as with any AI technology, careful consideration should be given to its broader deployment and potential misuse.

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

## A. Pipeline

As shown in Figure 8, our pipeline consists of two main training stages. (1) The SFT training stage first generates initial solutions that are validated through execution feedback, followed by critique generation where the generator learns to provide critiques based on execution feedback. These components are then used to train the final critic model through supervised finetuning. (2) The RL training stage leverages the critic's feedback to guide the generator in producing improved solutions, which are validated in a sandbox environment.

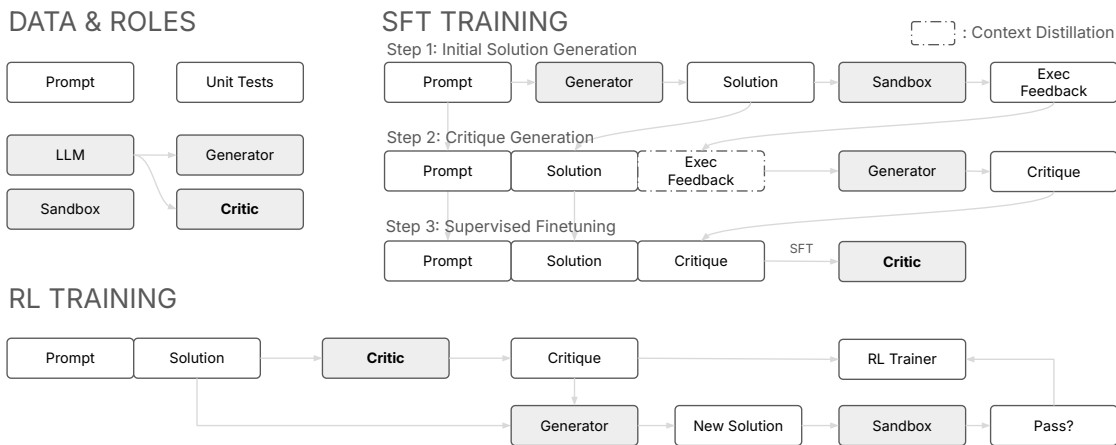

*Figure 8.* Overview of our two-stage training pipeline `CTRL`.

## B. Supplementary Discussion of Related Work

Table 6 categorizes prior methods into reward models, generative reward models, and critic models. Reward models like Standard RM (Bradley & Terry, 1952) and SynRM (Ye et al., 2024) focus on discrimination by outputting scalar rewards $r$ but lack refinement or critique supervision. Generative reward models, such as CLoud (Ankner et al., 2024) and Critic-RM (Yu et al., 2024), enhance discrimination by producing both rewards $r$ and critiques $c$, but their critiques primarily serve as a by-product for rewards rather than actionable refinement suggestions. Critic models, including UltraCM (Cui et al., 2023), Shepherd (Wang et al., 2023), and CriticGPT (McAleese et al., 2024), focus on generating critiques but rely heavily on human-annotated critique data, which limits scalability. In contrast, `CTRL` unifies discrimination and refinement by generating actionable critiques without direct supervision, leveraging execution feedback and reinforcement learning to enable scalable, iterative improvement.

*Table 6.* Comparison of reward models, generative reward models, and critic models.

| Methods | Input | Output | Discrimination | Refinement | Critique Supervision |
|---|---|---|---|---|---|
| Standard RM (Bradley & Terry, 1952) | $x, y$ | $r$ | ✓ | ✗ | ✗ |
| SynRM (Ye et al., 2024) | $x, y, c$ | $r$ | ✓ | ✗ | ✓ |
| UltraCM (Cui et al., 2023) | $x, y$ | $c$ | ✗ | ✓ | ✓ |
| Shepherd (Wang et al., 2023) | $x, y$ | $c$ | ✗ | ✓ | ✓ |
| CriticGPT (McAleese et al., 2024) | $x, y$ | $c$ | ✗ | ✓ | ✓ |
| CLoud (Ankner et al., 2024) | $x, y$ | $c, r$ | ✓ | ✗ | ✓ |
| Critic-RM (Yu et al., 2024) | $x, y$ | $c, r$ | ✓ | ✗ | ✗ |
| `CTRL` (Ours) | $x, y$ | $c$ | ✓ | ✓ | ✗ |

## C. Implementation Details

### C.1. Simulation

In our simulation (Section 2), we model the iterative refinement process using a Markov chain with parameters $p_{\text{init}}$, $p_{\text{cc}}$, and $p_{\text{cw}}$ to represent the initial correctness, the probability of maintaining correctness, and the probability of turning incorrect

solutions correct, respectively. Critiquing ability is controlled by varying $p_{cc}$ and $p_{cw}$ (e.g., strong critiquing: $p_{cc} = 0.9$, $p_{cw} = 0.3$; weak critiquing: $p_{cc} = 0.7$, $p_{cw} = 0.15$), while discrimination ability is adjusted via true positive rate (TPR) and false positive rate (FPR) (e.g., strong discrimination: $\text{TPR} = 0.7$, $\text{FPR} = 0.2$; weak discrimination: $\text{TPR} = 0.6$, $\text{FPR} = 0.3$). For each setting, we simulate $n$ refinement steps using Python, generating solutions based on refinement probabilities, applying a classifier to predict correctness, and selecting the best solution from predicted correct ones. The results are computed over 50,000 iterations and plotted to analyze the impact of critiquing and discrimination on final success rates. Specifically, the two processes — only using discrimination and using both discrimination and critiquing — are illustrated in Figure 9 to provide a clearer understanding of our simulation setup.

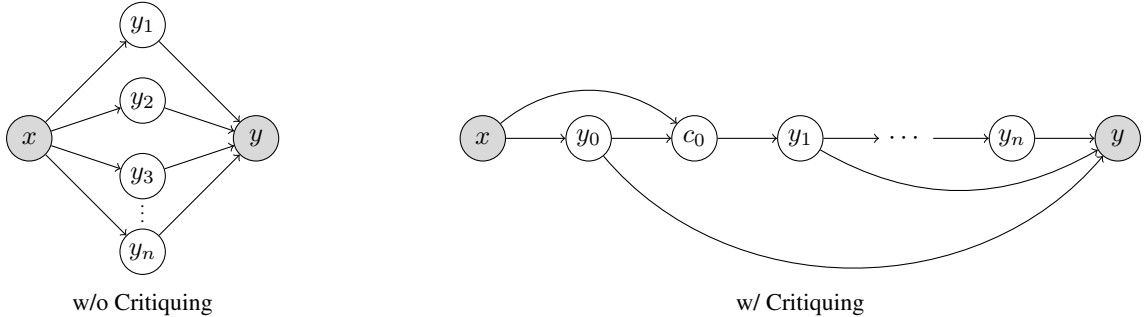

w/o Critiquing                                      w/ Critiquing

*Figure 9.* Graphical models for refinement processes: (left) only using discrimination (best-of-$n$ sampling) and (right) using both discrimination and critiquing (sequential critique-revision).

### C.2. Prompt Templates

**Critique-revision.**   The generator model $\pi(y \mid x)$ is implemented as a simple zero-shot generation process, where the model generates a solution $y$ directly from the problem statement $x$ without additional context or feedback. The critic model $C_\theta(c \mid x, y)$, as described in the main paper, generates textual feedback $c$ using a structured prompt that incorporates the problem $x$, the solution $y$, and explicit instructions to provide actionable and formatted suggestions. The improved solution distribution $\pi(y \mid x, y', c)$ is implemented as a two-turn process: in the first turn, the generator model drafts the initial solution $y'$ conditioned on the problem $x$ as the user message; in the second turn, the critique $c$ is presented as the user message, and the model revises the solution, conditioned on $x$, $y'$, and $c$.

**Execution-guided Critique Generation.**   To generate high-quality critiques (Section 3.2), we leverage execution feedback from a sandbox environment that evaluates the initial solution $y'$ against the test cases $T$ for the problem $x$. The execution results are mapped to predefined hint templates, which guide the critique generation process. The critic model is prompted with a structured template incorporating the problem $x$, the solution $y'$, and the corresponding hint $h$, enabling it to produce actionable and context-aware feedback. To prevent hallucination, critiques that explicitly reference the hints are filtered out. This ensures that the generated critiques are grounded in observable failures while effectively supporting solution refinement.

### C.3. Training

**Data Curation.**   Our data curation process starts with the TACO dataset (Li et al., 2023) and handles both function-based and input-output-based programming problems. We filter out malformed problems by removing those containing image tags and unusual HTML spans. For unit tests, we process them differently based on their type: function-based tests are converted to assertion statements, while input-output tests are standardized into a sandbox format with stdin-stdout pairs. We exclude problematic unit tests such as those with malformed string inputs (containing assignments or unexpected list operations) or invalid function calls. To avoid contamination, we further exclude 47 problems that overlap with our evaluation benchmarks. The final dataset is deduplicated based on problem descriptions, resulting in 18,820 problems.

**Supervised Finetuning.**   We leverage the synthesized critiques to perform supervised finetuning (SFT) on the model, enabling it to generate improved solutions. For each problem, we sample one initial solution and one corresponding synthesized critique, and train the model on these problem-solution-critique pairs. The training process follows the

---

**Prompt Template for Critique Generation**

```
You are tasked with analyzing an answer to a problem and providing constructive feedback. Do NOT provide
    ↪ direct solutions.

Problem description:
<problem>
{problem}
</problem>

Answer:
<answer>
{answer}
</answer>

Structure your response using the following format (without <format> tags):
<format>
Analysis:
{{Analysis}}

Improvement suggestions:
{{Suggestions}}

Overall judgment: {{Correct/Incorrect}}
</format>
```

---

**Prompt Template for Execution-guided Critique Generation**

```
You are tasked with analyzing an answer to a problem and providing constructive feedback. Do NOT provide
    ↪ direct solutions.
Please carefully reason about the hint to guide the user.
**Important: Do NOT mention 'the hint' in your feedback.**

Problem description:
<problem>
{problem}
</problem>

Answer:
<answer>
{solution}
</answer>

Hint:
<hint>
{hint}
</hint>

Structure your response using the following format (without <format> tags):
<format>
Analysis:
{{Analysis}}

Improvement suggestions:
{{Suggestions}}

Overall judgment: {{Correct/Incorrect}}
</format>
```

---

hyperparameters outlined in Table 8.

**RL Training.** We use VeRL (Sheng et al., 2024) as the codebase to optimize the model's generation quality. During RL training, we sample 4 initial solutions for each problem and train the critic model on all corresponding problem-solution pairs. This approach helps mitigate overfitting by exposing the critic to a diverse set of solutions for each problem. The RL training process follows the hyperparameters outlined in Table 9.

*Table 7.* Mapping between execution results and hint templates used for critique synthesis.

| Execution Result | Hint |
|---|---|
| Success (100%) | The draft solution is correct. A concise and positive feedback is recommended. |
| Failure (0%) | The draft solution is entirely wrong. A concise feedback requesting a fresh restart is recommended. |
| Partial Success | Input: {input}

Expected Output: {expected_output}

Actual Output: {actual_output} |
| Runtime Error | The code block:
```python
{code_block}
```

raised {error}. |

*Table 8.* SFT Hyperparameters.

| Parameter | Value |
|---|---|
| Learning Rate | $2 \times 10^{-5}$ |
| Learning Rate Schedule | Cosine |
| Training Batch Size | 256 |
| Maximum Token Length | 2,048 |
| Training Epochs | 1 |
| Mixed Precision Format | bfloat16 |

*Table 9.* RL Hyperparameters.

| Parameter | Value |
|---|---|
| Training Batch Size | 1,024 |
| Mini-Batch Size | 256 |
| Group Size | 8 |
| Learning Rate | $1 \times 10^{-5}$ |
| KL Coefficient | 0.001 |
| Maximum Prompt Length | 1,536 |
| Maximum Response Length | 768 |
| Temperature | 1.0 |
| Training Epochs | 2 |

## C.4. Evaluation.

**Inference.** During inference, we use a temperature of 0.7 for generating both initial solutions and critiques, while revised solutions are generated using greedy decoding. The maximum number of tokens generated is set to 1,024 for all stages.

**Reward Calculation.** To calculate rewards for our JudgeBench evaluation (Section 4.3), we use a critic model to assess the quality of solutions. Specifically, we generate multiple critiques for each solution and aggregate the results through majority voting. For each solution pair, the critic model compares the frequency of being labeled as "Correct" to determine which solution is better. As shown in Figure 10(a), we find that the accuracy of this majority voting strategy improves as the number of votes increases.

**Code Similarity Calculation.** To measure code similarity while accounting for semantically equivalent code with different variable names, we follow (Zheng et al., 2024) and implement a two-step comparison approach. We first normalize the code by parsing it into an Abstract Syntax Tree (AST), systematically renaming variables to canonical forms, and converting back to consistently formatted text. We then compute a similarity ratio using Python's `difflib.SequenceMatcher`, which represents the proportion of matching characters in the optimal alignment of the two normalized code sequences. This approach yields a score between 0 and 1, allowing us to identify structurally similar solutions regardless of variable naming choices.

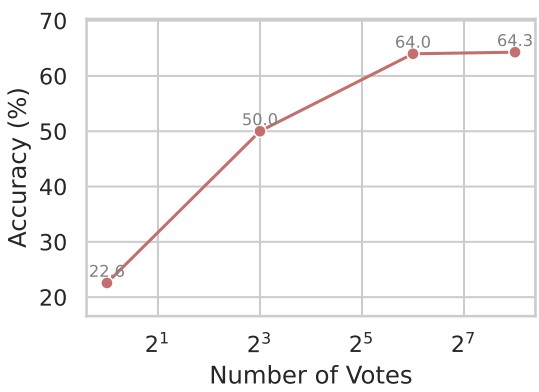

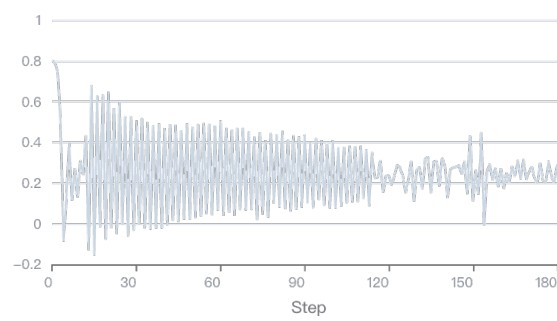

(a) The effect of the number of votes on the accuracy of majority voting in reward calculation. As the number of votes increases, the accuracy improves significantly, demonstrating the scalability and robustness of the majority voting approach.

(b) Training curve of the value network in PPO, showing the mean predicted value over training steps.

*Figure 10.* (a) Majority voting accuracy improvement with increasing votes. (b) Training curve of the value network.

## D. The Credit Assignment Issue for Critic Training

Our initial attempts to train critics using Proximal Policy Optimization (Schulman et al., 2017) for RL training revealed challenges in credit assignment for critique generation, as evidenced by the unstable value predictions shown in Figure 10(b). This difficulty arises because the quality of a critique is inherently holistic — it depends on multiple interdependent aspects like accuracy, completeness, and constructiveness — making it hard for the value network to learn which parts of the generated critique contributed to the final reward. These observations motivated our switch to GRPO, which circumvents the value prediction problem by using Monte Carlo sampling to directly estimate advantages, leading to more reliable credit assignment without the need for a potentially unstable value network.

## E. `CTRL` Samples

In Tables 10, 12 and 14, we present sample critiques generated by `CTRL` for solutions provided by Qwen2.5-Coder.

*Table 10.* A sample problem from CodeContests (Li et al., 2022), with by the solution generated by Qwen2.5-Coder and the critique provided by the CTRL critic.

| Problem & Solution | Critique |
|---|---|
| You had n positive integers a_1, a_2, ..., a_n arranged in a circle. For each pair of neighboring numbers (a_1 and a_2, a_2 and a_3, ..., a_{n - 1} and a_n, and a_n and a_1), you wrote down: are the numbers in the pair equal or not. | |
| Unfortunately, you've lost a piece of paper with the array a. Moreover, you are afraid that even information about equality of neighboring elements may be inconsistent. So, you are wondering: is there any array a which is consistent with information you have about equality or non-equality of corresponding pairs ? | Analysis:
The provided solution attempts to construct an array based on the sequence of 'E' and 'N' but does not correctly handle the circular nature of the array and the constraints imposed by these characters. Specifically, the approach does not ensure consistency across the entire circular arrangement, particularly at the boundary where the last element connects back to the first. |

```python
def can_form_array(s):
    n = len(s)
    # We will try to assign values to the array a
    # Start with assigning 1 to the first element
    a = [1] * n

    # Try to assign values based on the given
        conditions
    for i in range(n):
        if s[i] == 'E':
            a[(i + 1) % n] = a[i]
        else:
            a[(i + 1) % n] = a[i] + 1

    # Check if the last and first elements are
        consistent with s[n-1]
    if s[-1] == 'E' and a[0] != a[-1]:
        return "NO"
    if s[-1] == 'N' and a[0] == a[-1]:
        return "NO"

    return "YES"

def main():
    import sys
    input = sys.stdin.read().splitlines()
    t = int(input[0])
    results = []
    for i in range(1, t + 1):
        s = input[i]
        results.append(can_form_array(s))

    for result in results:
        print(result)

if __name__ == "__main__":
    main()
```

The code assigns values to the array elements based on whether the current character is 'E' or 'N', but it does not account for potential conflicts that arise when the array wraps around. This means that the final check between the first and last elements might fail due to inconsistencies introduced earlier in the array construction process.

Improvement suggestions:
1. Consider using a more systematic approach to handle the circular nature of the array. One possible method is to use a graph traversal algorithm, treating each position in the array as a node and each 'E' or 'N' as an edge with a label indicating equality or inequality.
2. Use a union-find data structure to manage equivalence classes of indices that must hold the same value due to 'E' connections.
3. Iterate through the string and apply the rules: if 'E', ensure the indices are in the same set; if 'N', ensure they are in different sets.
4. After processing the string, verify that no conflicts exist in the union-find structure, particularly focusing on the circular connection between the first and last elements.
5. If no conflicts are found, it is possible to construct an array that satisfies the conditions; otherwise, it is not.

Overall judgment: Incorrect

*Table 12.* A sample problem from LiveCodeBench (Jain et al., 2024), with by the solution generated by Qwen2.5-Coder and the critique provided by the CTRL critic.

| Problem & Solution | Critique |
|---|---|
| You are given a binary string s and an integer k. A binary string satisfies the k-constraint if either of the following conditions holds:

The number of 0's in the string is at most k.
The number of 1's in the string is at most k.

Return an integer denoting the number of substrings of s that satisfy the k-constraint. | Analysis:
The provided solution attempts to solve the problem by counting substrings based on the maximum occurrence of '0' and '1' up to `k`, but it does not correctly account for all possible substrings that satisfy the k-constraint. The approach incorrectly calculates the intersection of substrings that satisfy both conditions without properly considering the overlap between these substrings. Additionally, the logic for counting substrings is flawed as it does not consider every possible substring of `s`; instead, it focuses on contiguous sequences of '0's and '1's up to `k`. |

```python
class Solution:
    def countKConstraintSubstrings(self, s: str, k:
        int) -> int:
        def count_substrings_with_max_char(s,
            max_char, k):
            left = 0
            count = 0
            current_count = 0
            result = 0

            for right in range(len(s)):
                if s[right] == max_char:
                    current_count += 1

                while current_count > k:
                    if s[left] == max_char:
                        current_count -= 1
                    left += 1

                result += right - left + 1

            return result

        count_0s = count_substrings_with_max_char(s,
            '0', k)
        count_1s = count_substrings_with_max_char(s,
            '1', k)

        return count_0s + count_1s
```

Improvement suggestions:
1. **Re-evaluate the Approach**: Instead of focusing on contiguous sequences of '0's and '1's, the solution should consider every possible substring of `s` and check if it satisfies the k-constraint.
2. **Iterate Over All Substrings**: Use nested loops to iterate over all possible substrings of `s`. For each substring, count the occurrences of '0' and '1' and check if either count is less than or equal to `k`.
3. **Count Valid Substrings**: Maintain a counter to keep track of how many substrings meet the k-constraint criteria.
4. **Simplify the Logic**: Consider using Python's slicing capabilities to easily extract and evaluate each substring without unnecessary checks for contiguous characters.

Overall judgment: Incorrect

*Table 14.* A sample problem from MBPP+ (Liu et al., 2024a), with by the solution generated by Qwen2.5-Coder and the critique provided by the CTRL critic.

| Problem & Solution | Critique |
|---|---|
| Write a python function to check whether every odd index contains odd numbers of a given list.

```python
def odd_length_sum(arr):
    n = len(arr)
    res = 0

    # Iterate through each element in the array
    for i in range(n):
        # Calculate the number of subarrays in which
        #     arr[i] is present
        count = ((i + 1) * (n - i) + 1) // 2

        # If the count is odd, add the element to
        #     the result
        if count % 2 == 1:
            res += arr[i]

    return res
``` | Analysis:
The provided solution checks if the parity (odd/even nature) of the numbers at odd indices matches the index itself. This means it verifies that a number at an odd index is odd, and a number at an even index is even. However, this does not correctly address the problem at hand, which requires checking whether every odd index contains an odd number, not just matching the parity of the index.

Improvement suggestions:
- Review the logic used in the function. Instead of checking if the parity of the number matches the parity of the index, the function should directly check if the number at each odd index is odd.
- Consider using a more straightforward loop or list comprehension to achieve the desired result.

Overall judgment: Incorrect |

