# OpenReview forum: "Teaching Language Models to Critique via Reinforcement Learning"
_ICML.cc/2025/Conference — ICML 2025 poster_

### Official Review · Reviewer_aMwi · 2025-03-03

**Overall Recommendation:** 3

**Summary:**

This paper introduces CTRL (Critic Training via Reinforcement Learning), a framework designed to train critic models for iterative refinement in code generation tasks. The authors propose a two-stage pipeline: supervised fine-tuning (SFT) using execution-guided critique synthesis and reinforcement learning (RL) with Group Relative Policy Optimization (GRPO). CTRL decouples the critic model from the task-performing model, enabling it to provide actionable feedback that improves solution quality without human supervision. Experimental results on multiple programming benchmarks demonstrate significant improvements in pass rates, reduced compounding errors, and generalization capabilities across both base and stronger generator models.

**Claims And Evidence:**

The claims made in the paper are generally well-supported by experimental results. The authors demonstrate that CTRL-trained critics improve pass rates, reduce error compounding, and generalize across different generator models and benchmarks. However, some claims, such as the scalability of CTRL to broader tasks beyond code generation, are only indirectly supported and lack extensive empirical evidence. Additional validation on more diverse domains could strengthen these claims.

**Essential References Not Discussed:**

Some highly related works are not discussed:

1. CodeDPO: Aligning Code Models with Self Generated and Verified Source Code
2. Training Language Model to Critique with Multi-agent Feedback
3. RLEF: Grounding Code LLMs in Execution Feedback with Reinforcement Learning

**Experimental Designs Or Analyses:**

The experimental design is robust, with clear comparisons between CTRL and baseline methods. The use of multiple benchmarks and generator models adds credibility to the results. However, there are some limitations, such as the lack of ablation studies to isolate the contributions of each component (e.g., SFT vs. RL). Additionally, while the authors analyze error compounding and scalability, the experiments are primarily focused on code generation, limiting generalizability to other domains.

**Methods And Evaluation Criteria:**

The methods and evaluation criteria are appropriate for the problem. The use of programming benchmarks like CodeContests, LiveCodeBench, and MBPP+ ensures that the evaluation is rigorous and relevant to the task of code generation. The iterative critique-revision process and the reliance on execution feedback are well-aligned with the problem's requirements. However, the reliance on sandbox execution environments might limit scalability to tasks without such clear evaluation metrics.

**Other Comments Or Suggestions:**

No

**Other Strengths And Weaknesses:**

Strengths:

The paper presents a novel combination of supervised fine-tuning and reinforcement learning for training critics.
The experimental results demonstrate significant improvements over baselines, especially in reducing error compounding and enabling multi-turn critique-revision.
The framework's ability to generalize across generator models and tasks is a notable strength.

Weaknesses:

This paper only focus on the code generation task, leaving alone other tasks especially the open-domain tasks, like summarization, translation, alignment.

**Questions For Authors:**

No questions

**Relation To Broader Scientific Literature:**

The paper builds upon prior work in self-improvement of large language models, self-critique, and reinforcement learning for feedback generation. It extends these ideas by introducing a scalable framework that leverages execution feedback and RL. The weak-to-strong generalization phenomenon observed in CTRL aligns with findings in scalable oversight and weak supervision. However, the paper could better contextualize its contributions relative to recent advancements in generative reward models and self-correction in LLMs.

**Theoretical Claims:**

I did not verify all mathematical details rigorously, and some minor steps in the derivations (e.g., bias analysis in Equation 2) could benefit from additional clarification. While the claims are likely correct, their presentation could be more transparent for broader accessibility.

---

> ### Author Rebuttal · Authors · 2025-03-31
>
> Thank you for your thoughtful feedback! We appreciate your recognition of our paper’s strengths, especially the `novel combination` of supervised fine-tuning and reinforcement learning for training critics and the `robust experimental design` supporting our claims. Below, we address your specific concerns in detail:
>
> **Scalability to broader tasks beyond code generation**:
>
> We agree that demonstrating generalization beyond code generation would strengthen our paper. We have conducted **additional experiments** on IFEval [1], a benchmark commonly used for evaluating alignment capabilities [2]. Our results show that, while not explicitly trained on instruction-following tasks, CTRL improves performance on alignment tasks through iterative refinement, demonstrating our approach can generalize to tasks beyond code generation. We will incorporate these findings in our final version to better illustrate CTRL's broader applicability.
>
> [1] Zhou J, Lu T, Mishra S, et al. Instruction-following evaluation for large language models
>
> [2] Yang A, Yang B, Zhang B, et al. Qwen2. 5 technical report
>
> **Sandbox reliance of our method**:
>
> While we leverage sandbox execution during training, CTRL can be adapted to tasks without such verification tools through 1) learned reward models for tasks like safety; and 2) reference-based evaluation for tasks with reference responses (e.g., translation) using metrics like ROUGE or BLEU.
>
> In fact, our framework is agnostic to the specific reward mechanism as long as it can distinguish between successful and unsuccessful revisions.
>
> **Ablation of components (e.g., SFT vs. RL)**:
>
> Tables 1 and 2 provide an initial component analysis comparing CTRL-SFT and full CTRL performance. Together with our **additional experimental results** on Pass@5 performance in the [anonymous link](https://anonymous.4open.science/r/icml25-rebuttal/anonymous.pdf), Table 2, we observe that:
> - SFT improves discrimination (F1: 61.19% → 68.55%) and establishes the critique format
> - RL significantly improves single-turn critique-revision (Pass@1: 8.36% → 11.76%) while reducing regression ($\Delta_\downarrow$: 3.03% → 0.85%)
> - For multi-sample scenarios, SFT shows larger gains in Pass@5, while RL improves single-sample effectiveness
>
> This demonstrates that while SFT provides initial improvement in critique format and discrimination, RL training is crucial for generating feedback that leads to successful revisions.
>
> **Essential references not discussed**:
>
> We thank the reviewer for suggesting highly relevant papers we overlooked. We will include and discuss these works in our final version:
>
> - While CodeDPO improves code generation through DPO with self-generated validation, our work differs by specifically training a critic model to provide human-readable feedback. Besides, CodeDPO relies on larger models for data generation, whereas our approach requires only self-generated critiques for critic training.
> - MultiCritique shares our goal of training critic models but utilizes a multi-agent framework with GPT-4 for meta-critique. In contrast, CTRL achieves strong performance without requiring access to more powerful models during training. Besides, our method enjoys the simplicity of having one critic model instead of four different models.
> - While RLEF uses execution feedback to improve generator models directly, we focus on enhancing the feedback capability of LLMs. In this regard, our approaches are complementary - RLEF trains better generators, while CTRL trains better critics.
>
> **Contributions relative to recent advancements in generative reward models and self-correction**:
>
> We discuss our position relative to generative reward models in Appendix B, Table 6. Specifically, our approach uniquely **unifies discrimination and refinement** by generating actionable critiques **without direct human supervision**. Our work differs from recent self-correction approaches:
> - SCoRe [3] employs multi-turn online RL to improve self-correction, but focuses on the generator directly correcting its outputs. CTRL decouples critique and generation, allowing specialized training of the critic and demonstrating signs of scalable oversight.
>
> - GLoRe [4] decomposes refinement into when, where, and how to refine. However, they focus on training both global and local refinement models separately, while CTRL trains a specialized critic model that can identify errors and provide actionable feedback to any generator.
>
> [3] Kumar A, Zhuang V, Agarwal R, et al. Training language models to self-correct via reinforcement learning
>
> [4] Havrilla A, Raparthy S, Nalmpantis C, et al. Glore: When, where, and how to improve llm reasoning via global and local refinements
>
>
> We again appreciate your thoughtful review and will incorporate your suggestions to strengthen the final version of our paper.

---

### Official Review · Reviewer_VcEc · 2025-03-13

**Overall Recommendation:** 3

**Summary:**

This paper presents the CTRL framework - a two-stage training approach that separates the critique function of a language model from its generative capabilities. The authors first synthesize high-quality critiques using execution feedback (running code trying to go through unit tests), which are then used in a supervised fine-tuning phase of a dedicated critique model. In the second phase, they use Group Relative Policy Optimization (GRPO) to further refine the critics through reinforcement learning so that the generated critiques directly help the fixed generator model to improve its output (measured in terms of test pass rates). Experiments on multiple coding benchmarks (e.g., CodeContests, LiveCodeBench, MBPP+) and JudgeBench have shown dramatic improvements in the performance of the framework, with reported relative improvements of up to 106% on some metrics. The framework not only improves the pass rate of the base generator, but also generalizes well when applying the critic to stronger models (e.g., GPT-4). In summary, this paper presents a novel, mathematically-based, empirically validated method for guiding LMs to provide actionable critiques that significantly improve the correctness of generated code.

**Claims And Evidence:**

Yes

**Essential References Not Discussed:**

No

**Experimental Designs Or Analyses:**

Yes, no further issues

**Methods And Evaluation Criteria:**

Yes

**Other Comments Or Suggestions:**

- Authors should discuss the possibility of applying CTRL to domains where feedback is less binary or difficult to obtain (e.g., essay writing, dialogue generation).
Future work could explore how to build alternative feedback mechanisms in these domains.

- It would be useful to discuss the sensitivity of the method to various hyperparameters (e.g., group size in GRPO, strength of KL regularization) and whether any parts of it can be simplified.

- Providing more qualitative examples, and perhaps a manual evaluation of the generated critiques, would help provide insight into what strategies the critics have learned and whether these critiques are interpretable.

**Other Strengths And Weaknesses:**

## Strengths
- Solid mathematical and theoretical foundation:
The iterative improvement process is described through Markov chain modeling, which clearly illustrates the impact of criticism and discrimination ability on the final success rate.
Normalization of the strategy gradient using GRPO significantly reduces the gradient variance, thus making the reinforcement learning process stable and efficient.
- Experimental validation is adequate:
Significant improvement is demonstrated on several code generation benchmarks (CodeContests, LiveCodeBench, MBPP+), and the critic's discriminative ability is also verified on JudgeBench.
Experiments comparing self-criticism, raw generation, and improvements under different generation models show that CTRL effectively reduces error accumulation and dramatically improves pass rates

## Weaknesses

- Methods are not innovative enough:
Although CTRL has done some work in combining executive feedback, supervised fine-tuning and reinforcement learning, it has overall borrowed existing ideas and techniques, such as self-criticism, RLHF, and GRPO methods.
The core idea of the method is mainly to combine existing techniques, and it does not propose a completely new mechanism in terms of theory or algorithm, so it may not reach the highest level of innovation.
- The theoretical analysis is not deep enough:
Although relevant RL objectives and GRPO updating strategies are given, there is a lack of theoretical proof of convergence and optimality of the RL part, and the approach relies more on experimental performance.

**Questions For Authors:**

- How do you think about the CTRL framework being applied to automate the evaluation of tasks that are not as simple as code ?

- Can you describe the sensitivity of your method to the parameters in GRPO? Do you observe any significant performance degradation if these parameters are changed?

- In addition to improving the pass rate, have you performed a qualitative assessment of the quality of the generated comments?
For example, how well do these comments agree with human debugging strategies?

- How robust is your approach if execution feedback (i.e., test cases) is noisy or incomplete?
Have you tested situations where the feedback could be misleading?

**Relation To Broader Scientific Literature:**

The paper's key contributions extend prior work on LLM self-improvement by addressing critical limitations identified in existing literature. While earlier methods like self-critique (Madaan et al., 2024) showed theoretical promise for iterative refinement, Huang et al. (2023) demonstrated their practical limitations due to compounding errors: a challenge CTRL directly addresses through its specialized critic training. The framework builds on weak-to-strong generalization concepts (Christiano et al., 2018) but extends them to cross-model critique-revision dynamics, showing smaller critics can effectively guide larger generators. This advances beyond traditional reward modeling approaches (Gao et al., 2023) that used scalar feedback, instead aligning with emerging work on generative reward models (Yu et al., 2024) while eliminating their reliance on human annotations. The test-time scaling mechanism through iterative revisions responds to Snell et al.'s (2024) call for compute-efficient inference methods, but introduces novel critique-driven iteration rather than simple sampling. By formalizing the critique process through Markov chain analysis, the work provides theoretical grounding to empirical observations in code generation studies (Zheng et al., 2024) about the importance of actionable feedback, while offering a generalizable framework that could extend beyond programming domains.

**Theoretical Claims:**

The paper makes reasonable empirical claims about critic training but lacks formal theoretical proofs for its Markov chain analysis and optimization properties. The most significant theoretical gaps are:

Missing justification for Markov assumptions in iterative refinement
Unquantified variance reduction claims for GRPO
Unexamined optimization landscape characteristics
These limitations don't invalidate the experimental results but suggest opportunities for deeper theoretical analysis in future work. The community would benefit from formal proofs of:

Markov chain convergence properties
GRPO variance bounds
Conditions enabling weak critic supervision of strong generators

---

> ### Author Rebuttal · Authors · 2025-04-01
>
> Thank you for your comprehensive and thoughtful review! We appreciate your recognition of our work's `solid mathematical and theoretical foundation` and `adequate experimental validation.` We would like to address your concerns in detail:
>
> **Theoretical analysis**:
>
> Our work is primarily empirical. While we use mathematical frameworks to motivate our approach, our main contributions are demonstrating a practical framework for critic training and empirically validating its effectiveness.
>
> To strengthen our theoretical analyses, we discuss more in detail:
>
> - Markov chain analysis: We provide theoretical analyses of Markov chain convergence properties in the [anonymous link](https://anonymous.4open.science/r/icml25-rebuttal/anonymous.pdf) complementing Figure 3's empirical findings.
> - GRPO properties: We show that GRPO achieves variance reduction by a factor of $G$, explaining its stability with large critique spaces. Our analysis also demonstrates that GRPO ensures monotonic improvement and converges to a local optimum under standard assumptions.
> - Weak-to-strong generalization: Our findings that weaker critics can guide stronger generators align with recent work (Burns et al., 2023; Kenton et al., 2024), though theoretical conditions for when this works remain an open research question.
>
> **CTRL does reach the highest level of innovation**:
>
> Thank you for recognizing our `novel, mathematically-based, empirically validated method.` Our primary contributions include:
>
> - Problem formulation: We identify and formalize the critique generation problem through a Markov chain lens, distinguishing between discrimination and critiquing abilities.
> - System design: We propose a decoupled critic-generator architecture and a two-stage training pipeline specifically designed for critique learning.
> - Robust training: We demonstrate GRPO's suitability for critique training due to reduced variance in policy gradients, and demonstrate its effectiveness empirically.
> - Empirical findings: We provide extensive empirical evidence demonstrating that our framework enables test-time scaling, mitigates compounding errors, and facilitates weak-to-strong generalization.
>
> **Applying CTRL to domains where feedback is less binary**:
>
> For domains where feedback is less binary, CTRL can leverage alternative reward mechanisms:
>
> - Learned reward models: Training a reward model from pairwise preference data (as in RLHF approaches) for tasks like safety or preference alignment.
> - Reference-based evaluation: For tasks with reference responses (e.g., translation, summarization), we can use metrics like log-likelihood on reference outputs or automated evaluation metrics (e.g., ROUGE and BLEU) as reward signals.
>
> CTRL is agnostic to the specific reward mechanism as long as it can distinguish between successful and unsuccessful revisions. This flexibility extends CTRL beyond domains with clear verification tools.
>
> Notably, we observe positive generalization results on IFEval, a benchmark outside our training distribution, as shown in our anonymous link.
>
> **Sensitivity to GRPO parameters**:
>
> We relied on prior work in RL training that identified the most sensitive parameters for training stability. Due to computation constraints, we performed a limited grid search (<10 runs), with final parameter selection based on performance on a TACO test subset. We found our method relatively robust to GRPO hyperparameters within reasonable ranges:
>
> - Group size: Larger sizes improve performance (we used 8) but increase computation time
> - KL coefficient: While unnecessary, KL improves stability and KL=0.001 balances exploration and stability  in our experiments.
> - Learning rate: Most sensitive parameter; we found 1e-5 to be optimal via the grid search.
>
> **Qualitative assessment of critiques**:
>
> To better assess the quality of the generated critiques,  we manually evaluated critiques on 50 randomly sampled critiques from CTRL against Qwen2.5-Coder's solutions on CodeContests. Our analysis revealed CTRL employs diverse debugging strategies: algorithm improvement (43 instances), static analysis (38), strategic debugging (11), and dynamic testing (10).
>
> This distribution reveals that CTRL primarily focuses on structural code improvements and algorithmic enhancements rather than superficial issues. The prevalence of static analysis and refactoring strategies also suggests CTRL learns debugging approaches that align with human reasoning patterns.
>
>
> **Robustness to noisy feedback**:
>
> While we didn't explicitly test noisy feedback scenarios, our low regression rates ($\Delta_\downarrow$) demonstrate CTRL can distinguish correct implementations from problematic code, avoiding misleading critiques that worsen solutions. This inherently suggests robustness to noise, though we agree more systematic evaluation of noise tolerance is an important direction for future work.
>
> Thank you again for your detailed review. We hope our response addresses your concerns and questions.

---

> > ### Comment · Reviewer_VcEc · 2025-04-05
> >
> > Thank you for the response. Most of my concerns have been adequately addressed, and I have raised my score. Although I still find the novelty somewhat limited, I will vote for a weak accept.

---

> > > ### Author Response · Authors · 2025-04-06
> > >
> > > Thank you for reconsidering our work! We are glad that our response has adequately addressed your concerns and remain open to any further questions or clarifications you might need.

---

### Official Review · Reviewer_MYma · 2025-03-13

**Overall Recommendation:** 3

**Summary:**

The paper presents a method to teach an LLM to critique the response of another LLM, specifically in the domain of coding contest problems. The problem is formalized as maximizing the probability of the latter LLM to succeed at providing a correct response after seeing a critique produced by the former, which is an RL problem.

The method consists of an initial SFT stage followed by an RL stage. In the SFT stage, the critique model is trained on filtered critiques produced using a method that includes execution of the code produced by the answer model. In the RL stage the critique model is trained using GRPO to optimize the success of the answer model.

The results show a significant boost in performance on CodeContests, and MBPP, and to a lesser extent on LiveCodeBench. The performance further increases with multiple rounds of critique and revision (even though this was not done during training).

**Claims And Evidence:**

The main claim of the paper, that one can train a model to critique using RL (and SFT), is well supported by the experiments. The motivating ideas and claims made in section 2 and 3 are very well supported by the cited related works.

**Essential References Not Discussed:**

no

**Experimental Designs Or Analyses:**

The paper trains on the TACO dataset and evaluates on CodeContests, but these contain many of the same problems. The appendix states that 47 problems were excluded, but the TACO paper explicitly mentions that about half the TACO problems are also in CC, so I consider this a no-go. Moreover, the results in Table 3 show that although CTRL performs well on CodeContests and MBPP+ (similar contamination risk), on LiveCodeBench it fares worse than using either GPT-4o or Qwen2.5-Coder as critique models. This suggests that the model has not learned generalizable critiquing strategies that work beyond the narrow (and contaminated) distribution of TACO/CC.

Although this is a serious issue that must be discussed in more detail in the paper and acknowledged as a weakness, it is still interesting to see that the model succeeds in-domain. It is quite likely that drastically scaling up the data size and diversity could enable learning of generalizing critiquing policies.

**Methods And Evaluation Criteria:**

The method is rather straightforward and sensible. Similarly the evaluation datasets and benchmarks are standard (though see comments on experimental design below)

**Other Comments Or Suggestions:**

- I would not call the critic model Q, since this has a well-established meaning in RL
- typo on page 5, "we samples", "and computes"
- The equation for J on page 5 shows Q(y|x) but I think it should be Q(c | x, y)?

**Other Strengths And Weaknesses:**

The experimental section of the paper contains a number of interesting analyses.

**Questions For Authors:**

n/a

**Relation To Broader Scientific Literature:**

The paper very clearly motivates its approach by citing relevant related work. This is a very nice feature of the paper.

**Theoretical Claims:**

The claim on page 5 that variance of the gradient scales with the size of the answer and critique space are not proven. Other than that the paper does not make significant theoretical claims.

---

> ### Author Rebuttal · Authors · 2025-03-31
>
> Thank you for your detailed review and valuable feedback! We appreciate your recognition that our main claim `is well supported by the experiments` with `a number of interesting analyses` and that the paper `very clearly motivates its approach by citing relevant related work.` We address your primary concerns below:
>
> **Proof: Variance of the gradient**:
>
> We provide a short proof in our [anonymous link](https://anonymous.4open.science/r/icml25-rebuttal/anonymous.pdf) that justifies our claim about gradient variance scaling with $|\mathcal{Y}| \cdot |\mathcal{C}|$.
> The key insights from this proof are:
> - When analyzing the variance of our policy gradient estimator, we find that under mild assumptions it scales proportionally with the product of answer and critique space sizes ($|\mathcal{Y}| \cdot |\mathcal{C}|$).
> - In cases where covariance terms are significant, the scaling becomes even worse than linear, potentially leading to a super-linear increase in the overall variance.
>
>
> This theoretical analysis provides a mathematical foundation for why standard policy gradient approaches struggle with critique-revision optimization, directly motivating our CTRL design choices that address this variance challenge. We will incorporate the proof in our final version.
>
>
>
> **Dataset overlap between TACO and CodeContests**:
>
> Thank you for raising this question! We agree that while we discuss this issue in the paper and propose a decontamination method to mitigate this, it deserves more attention. We would like to clarify a few points:
>
> - **Actual overlap**: While the TACO paper mentions a significant overlap with CodeContests overall, the critical distinction is that we evaluate on CodeContests test set (165 problems), while the reported overlap primarily exists between TACO and CodeContests full set. We excluded the 47 problems we identified as direct overlaps between the TACO train set and the CodeContests test set.
>
> - **Performance on out-of-distribution benchmarks**: We would like to highlight that our method demonstrates significant improvement over the zero-shot baseline on LiveCodeBench with Qwen2.5-Coder as the generator model (30.54% -> 33.21% in Pass@1), outperforming GPT-4o and Qwen2.5-Coder as critic models. This demonstrates that our method does learn useful critiquing abilities that transfer beyond its training distribution.
>
> - **Additional experiment**: To further validate CTRL's generalizability, we have conducted **additional experiments** on IFEval [1], a benchmark for instruction-following capabilities. The results are presented in the anonymous link, Table 1. Despite not being trained on instruction-following data, CTRL boosts accuracy by over 2% for single-turn critique-revision with Qwen2.5-Coder as the generator, outperforming all the baselines including GPT-4o.
>
> We agree that scaling up the data size and diversity is a promising direction to further improve the results. and consider this an important direction for future work, which we will highlight in the final version.
>
> [1] Zhou J, Lu T, Mishra S, et al. Instruction-following evaluation for large language models[J]. arXiv preprint arXiv:2311.07911, 2023.
>
>
> **Typos and suggestions**:
> Thank you for your careful review! We appreciate your suggestion about the naming convention, and will rename $Q$ and fix the spotted typos in the final version.
>
> Thank you again for your thorough review. Your feedback has helped us strengthen our work and identify important areas for improvement.

---

### Official Review · Reviewer_25hb · 2025-03-13

**Overall Recommendation:** 4

**Summary:**

Teaching large language models (LLMs) to critique and refine their outputs is crucial for building systems that can iteratively improve, yet it
is fundamentally limited by the ability to provide accurate judgments and actionable suggestions. In this work, the authors study LLM critics for code generation and propose CTRL, a framework for Critic Training via Reinforcement Learning, which trains a critic model to generate feedback that maximizes correction performance for a fixed generator model without human supervision. Their results demonstrate that critics trained with CTRL significantly enhance pass rates and mitigate compounding errors across both base and stronger generator models.

# Update after rebuttal
This paper is novel and clear in presentation, and the rebuttal resolves my concerns. I will keep my positive rating as my final score after rebuttal phase.

**Claims And Evidence:**

I summarize the claims issued in the paper, which includes:
- Challenge: Without appropriate external feedback, such self-improvement loops may lead to performance degradation.
- Solution 1: Reward models: reward models compress complex evaluation criteria into simplified numerical signals
- Solution 2: Automated verification tools: generate low-level execution traces that do not directly translate to high-level fixes
- Important: Feedback needs to both accurately discriminate the correctness of solutions and provide informative yet actionable suggestions for improvement.

All the claims align with my previous knowledge and have clear evidence in related works.

**Essential References Not Discussed:**

N/A

**Experimental Designs Or Analyses:**

The experimental design is abundant and comprehensive. But I have a major concern about the cost:
1. For reward evaluation, the authors use majority of vote for each step. Specifically, they generate multiple critiques for each solution and aggregate the results through majority voting. What is the base critic and the count is from 2 to $2^7$, does that cost too much?

2. We know that fine-tuning Qwen2.5-32B is costly with SFT and even more costly with RL, how the authors have such computing resource to afford fully fine-tuning with knowledge distillation from self-critique with execution feedback or RL training to optimize the GRPO goal with multiple critics?

**Methods And Evaluation Criteria:**

Yes, the critique qualtiy is important for underlying task performing model. Through (1 supervised fine-tuning to format the answer and enhandle the discrimination ability with hint, and (2 further RL training the critique through GRPO with advantage normalized reward that is calculated by majority voting of additional critique fixed evaluation models, they propose CTRL framework which is Qwen-2.5 based critique that surpasses multiple competitors like GPT-4o. And the experiment results and additional analysis about compounding errors, Test-time Scaling and Evaluating Critics as Generative Reward Models demonstrate the effectiveness of approach.

**Other Comments Or Suggestions:**

The reward calculate is too costly, could the authors just use the sandbox outputs as the correctness signal to save time?

**Other Strengths And Weaknesses:**

Strength:
The idea is novel and useful to improve the performance with solid experimental demonstration and analysis disucssion.
Motivation is clear, every step has a clear purpose.

Weakness:
- The implementation of methodology and baseline comparison requires large computing resources and long-time RL training with sensitive hyper-parameter tuning, how the authors overcome the difficulty and conduct hyper-parameter search for such a complex system with many components is interesting to know.

**Questions For Authors:**

Please see weakness part in "Other Strengths And Weaknesses" section.

**Relation To Broader Scientific Literature:**

This work relates to LLM critics for code generation, which applies GRPO to reduce variance through advantage and ensembles and chain of thoughts based prompt engineering with external ground-truth assistance (execution feedback) to improve the code generation quality. We note that such critique model can also serve as generative reward model for other domain tasks in JudgeBench.

**Theoretical Claims:**

N/A

---

> ### Author Rebuttal · Authors · 2025-03-31
>
> Thank you for your comprehensive and well-thought-out review! We appreciate your recognition of our work's novelty, effectiveness, experimental design, clear motivation, and solid supplementary material. Your positive feedback is truly encouraging.
>
> Below, we address your concerns about computational costs and system complexity:
>
> **Cost for majority voting in JudgeBench evaluation**:
>
> The base critic used for majority voting is our CTRL-trained Qwen2.5-Coder-32B model, the one used throughout our evaluation. The majority voting approach is primarily used for our JudgeBench evaluation (Section 4.3) to calculate pairwise accuracy.
>
> While generating multiple critiques may seem costly, modern inference engines significantly reduce this overhead through prefix caching (i.e., the same prompt is prefilled for generating multiple critiques). This majority voting approach is also used in recent work on generative verifiers [Zhang et al., 2024], which observe similar results to ours that majority voting boosts accuracy significantly.
>
> Our experiments show that performance plateaus after approximately $2^6$ critiques, with additional samples providing diminishing returns (64.3 at Maj@$2^7$ vs 64.0 at Maj@$2^6$). For cost comparison with Claude 3.5 Sonnet (which performs similarly on JudgeBench), we calculate per-problem inference costs using [Together AI pricing](https://www.together.ai/pricing) for Qwen2.5-Coder-32B and [Anthropic pricing](https://www.anthropic.com/pricing#anthropic-api) for Claude 3.5 Sonnet:
>
> CTRL with Maj@$2^6$:
> - Average prompt length: 2,067.17 tokens
> - Average response length per critique: 231.72 tokens
> - Input/output cost: $0.80/1M tokens
> - **Cost**: (2067.17 + 231.72 * 64) * 2 * 0.8 / 1M = $0.027 per problem
>
> Claude 3.5 Sonnet:
> - Average prompt length: 1,627.74 tokens
> - Average response length: 652.97 tokens
> - Input cost: $3/1M tokens
> - Output cost: $15/1M tokens
> - **Cost**: 1627.74 * 2 * 3 / 1M + 652.97 * 2 * 15 / 1M = $0.029 per problem
>
> Note that the factor of 2 accounts for calculation on both responses in CTRL and removing positional bias for Claude 3.5 Sonnet.
>
> This analysis demonstrates that our method achieves comparable performance to state-of-the-art proprietary models at a similar or slightly lower cost.
>
> **Cost for fine-tuning**:
>
> We acknowledge that RL training is computationally intensive, especially for large models. To manage this, we implemented several optimizations:
> - SFT stage: Our SFT on self-critique with execution feedback provides a strong initialization, significantly reducing the need for extensive RL exploration.
> - Optimization techniques: We leveraged fully sharded data parallelism (FSDP), gradient checkpointing, and sequence packing to reduce memory requirements and speed up training. At each step the critic model and the generator model are offloaded to CPU to save memory, allowing us to train 32B models with a minimum of 8 x 80G GPUs.
>
> As open-source RL training frameworks [1,2] continue to evolve, we expect these costs to decrease further in the future.
>
> [1] Sheng G, Zhang C, Ye Z, et al. Hybridflow: A flexible and efficient rlhf framework[J]. arXiv preprint arXiv:2409.19256, 2024.
>
> [2] von Werra L, Belkada Y, Tunstall L, et al. TRL: Transformer Reinforcement Learning[CP/OL]. GitHub repository. GitHub, 2020. https://github.com/huggingface/trl.
>
> **Clarification on "multiple critics"**:
>
> We want to clarify that our approach uses only a single critic model during training, not multiple critics as may have been implied. We sample multiple critiques from this single model to estimate advantages in GRPO.
>
> **Hyperparameter tuning**:
>
> For hyperparameter selection, we relied on prior work in RL training that identified the most sensitive parameters for training stability: learning rate, KL coefficient, and group size for GRPO. Due to computation constraints, we performed a very limited grid search over these key parameters (< 10 runs), with final parameter selection based on performance on a TACO test subset. We report the final hyperparameters in Table 9.
>
> We agree that more extensive hyperparameter search could potentially improve results. We will improve the final version of our paper with a more detailed description of our grid search process.
>
> **Sandbox outputs as correctness signal**:
>
> Regarding using sandbox outputs as correctness signals, we would like to clarify two contexts of “reward calculation”:
> - During CTRL training: We do use sandbox outputs (pass/fail) as the reward signal to train our critic model, as described in Section 3.1.
> - For JudgeBench evaluation: High-quality unit tests are not available for many tasks, especially general-domain questions. This practical limitation is precisely why CTRL is valuable - it internalizes sandbox signals during training, allowing critic models to generalize to new domains where explicit test cases are not available.
>
> Thank you again for your insightful review and positive assessment of our work.

---

### Decision · Program_Chairs · 2025-05-01

**Decision:**

Accept (poster)

**Comment:**

This paper proposes CTRL (critic training via reinforce learning) for code generation, a technique that learns a critique model and performs iterative refinement when paired with a generation model. To train the model, this work proposes two stages, the SFT stage an RL stage with the GRPO algorithm. The reviewers agree that the method is shown to yield solid improvements with comprehensive experiments. There are some concerns around the train/test overlap with the use of the TACO dataset, as well as detailed ablations on the improvements from the SFT and RL stages, which are adequately resolved during the rebuttal phase. However, the main remaining shared concern of this work is the lack of novelty, as methods like iterative refinement, SFT + RL (GRPO) tuning, and learning of critic models are all well-known methods not only for code generation but for many other tasks. Adding more discussions of the missing related works that the reviewers mentioned can also better help position this paper.